**www.cambridge.org/qrd**

# Diverse single-stranded nucleic acid binding proteins enable both stable protection and rapid exchange required for biological function

## Michael Morse[1], Ben A. Cashen[1], Ioulia Rouzina[2] and Mark C. Williams[1]

[1]Department of Physics, Northeastern University, Boston, MA, USA and [2]Department of Chemistry and Biochemistry, Ohio State University, Columbus, OH, USA

## Research Article

optical tweezers; protein–nucleic acid interactions; single molecule biophysics

**Corresponding author:**
Mark C. Williams;
Email: ma.williams@northeastern.edu

### Abstract

Single-stranded nucleic acid (ssNA) binding proteins must both stably protect ssNA transiently exposed during replication and other NA transactions, and also rapidly reorganize and dissociate to allow further NA processing. How these seemingly opposing functions can coexist has been recently elucidated by optical tweezers (OT) experiments that isolate and manipulate single long ssNA molecules to measure conformation in real time. The effective length of an ssNA substrate held at fixed tension is altered upon protein binding, enabling quantification of both the structure and kinetics of protein–NA interactions. When proteins exhibit multiple binding states, however, OT measurements may produce difficult to analyze signals including non-monotonic response to free protein concentration and convolution of multiple fundamental rates. In this review we compare single-molecule experiments with three proteins of vastly different structure and origin that exhibit similar ssNA interactions. These results are consistent with a general model in which protein oligomers containing multiple binding interfaces switch conformations to adjust protein:NA stoichiometry. These characteristics allow a finite number of proteins to protect long ssNA regions by maximizing protein–ssNA contacts while also providing a pathway with reduced energetic barriers to reorganization and eventual protein displacement when these ssNA regions are diminished.

## Introduction

### Optical tweezers force spectroscopy

Optical tweezers (OT) can isolate single nucleic acid (NA) molecules in suspension by tethering both ends to functionalized micron-sized beads which are held by laser trap(s). The system directly controls the end-to-end extension of the NA molecule based on the relative displacement of the two trapped beads while simultaneously measuring applied force based on the deflection of the trapping laser as the beads shift from the center of the trap. One method to measure the properties of the NA molecule is to produce a force–extension curve (FEC), in which the force response is measured over a wide range of extensions (Figure. 1a), typically by increasing or decreasing the extension in fixed steps at a continuous rate starting from low or high extensions, respectively. Both DNA and RNA are well represented by polymer chain models, with double-stranded (ds) and single-stranded (ss) polymers having very different properties. While dsNA is modeled as an extensible worm-like chain (WLC) (Baumann et al., 1997; Odijk, 2002), the end-to-end extension ($X$) of ssNA as a function of applied force ($F$) is best modeled as a freely jointed chain (Smith et al., 1996):

$$X(F) = L\left(\coth\left(\frac{2PF}{k_{\mathrm{B}}T}\right)\right)\left(1 + \frac{F}{S}\right) \tag{1}$$

Both ssDNA and ssRNA have a contour length ($L$) of ~0.56 nm/nt and a persistence length ($P$) of ~0.75 nm, which reflect the length and flexibility, respectively, of the sugar-phosphate backbone (Figure 1a). The backbone can elastically stretch as well, with elongation linearly proportional to applied force, with the elastic modulus (S, ~600 pN for ssDNA and ssRNA) indicating the force required to double the length of the substrate. Thus, the FEC has two main regimes: at low force, the FEC gently curves upwards as the ssNA molecule is straightened, approaching its contour length, and at high force, the extension continues to increase linearly past the contour length. Note that the FJC does not account for secondary structure, which can form at low force. Structures such as large defined hairpins found in certain biological systems can persist at higher forces. OT can be used to study specific interactions between these sequences and binding proteins, but here we focus on long (>1 knt), 50% GC content, substrates without stable secondary structure (>5 pN applied tension), in order to focus on non-sequence specific interactions between ssNA and proteins. That is, we average over the behavior of many sequences

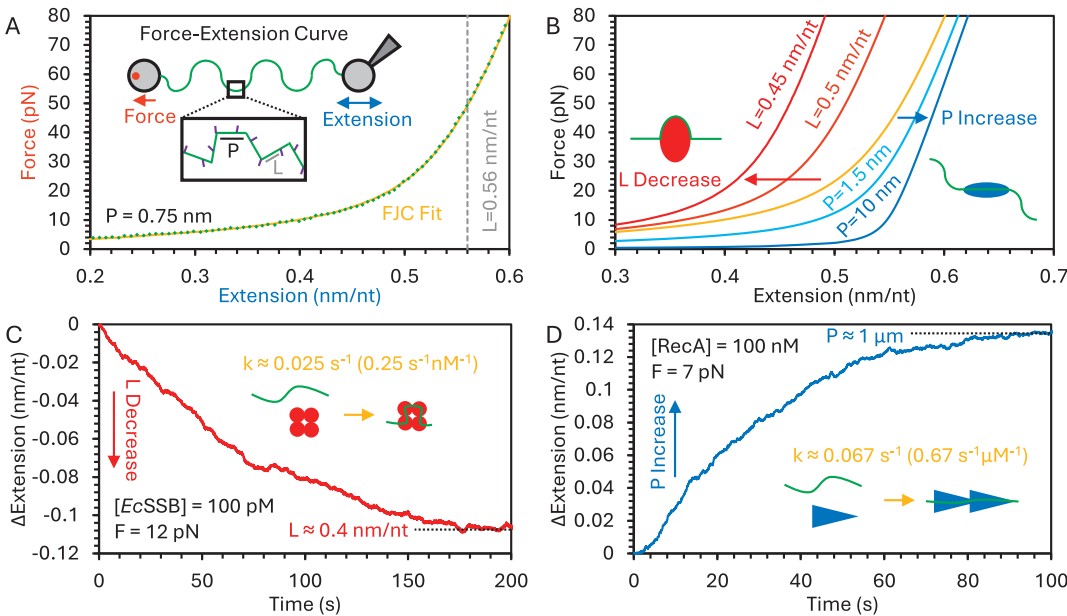

**Figure 1.** OT measurement of ssDNA conformation. (a) A single ssDNA molecule is tethered between two beads in an OT system to simultaneously measure end-to-end extension (blue) and applied force (red), generating an FEC (green). The data is fit by the FJC model (yellow), yielding the effective persistence and contour lengths. (b) The binding of proteins to the ssDNA can lower the contour length or increase the persistence length, resulting in a decrease or increase in extension, respectively, for a given applied force. (c) ssDNA bound by *Ec*SSB is wrapped around the protein homotetramer, effectively shortening its contour length. (d) RecA forms long, semi-rigid filaments along ssDNA, causing an increase in persistence length. Extension changes measured for ssDNA held at fixed tension in a protein reservoir are monotonic, exponentially approaching an equilibrium value at a rate governed by the concentration-dependent bimolecular NA–protein interaction.

present on the substrate, as the proteins discussed below also must interact with a wide range of sequences to coat exposed ssNA, as opposed to interacting only with specific sequences or motifs.

The polymer properties of the ssNA molecule determine the shape of its FEC, and if its properties change, so too will its FEC. For instance, an increase in persistence length both lowers the force required to straighten the ssNA and sharpens the transition between the low-force entropic regime and the high-force elastic regime (Figure 1*b*). In comparison, a reduction in contour length proportionally decreases ssNA extension at all forces, since extension scales linearly with contour length. In particular, the binding of a protein (or other small molecule) to ssNA can drastically impact its conformation. During binding, a length of ssNA adheres to a binding surface on the protein, such that its conformation is determined by the structure of the NA–protein complex. This has two primary effects. First, if the binding interface is not straight, bound ssNA must bend to adhere to the protein, which effectively lowers the substrate's contour length (Figure 1*c*). This effect is more pronounced for more circuitous paths through the binding surface, such as ssNA fully wrapping around the protein. Since ssDNA extension is directly proportional to contour length (eq. (1)), a consistent decrease in ssDNA extension is observed for all forces in the FEC. Second, since the binding interface is nearly always longer than the persistence length of bare ssNA (~0.75 nm or less than 2 nt), bound protein increases the effective persistence length (Figure 2*c*). The new effective persistence length can be determined by the number of nt bound by each individual protein or can become even longer if multiple proteins form a filament that remains rigid over many repeating protein subunits. ssDNA extension has a nonlinear response to a change in persistence length (eq. (1)), such that increased persistence length greatly increases ssDNA extension at low force but only mildly increases extension at high force when ssDNA is mostly straightened.

Proteins binding to ssDNA typically display both these effects, such that both effects can be observed in a single FEC, with increased persistence length increasing extension at low force but decreased contour length decreasing extension at high force (Cashen et al., 2023; Morse et al., 2019). However, applied force on the ssDNA may also impact the binding affinity and/or binding conformation of proteins, as tension inhibits wrapping.

In either case, when a ssNA substrate is incubated with a protein that has such an effect when bound, the extension change over time has the shape of exponential decay, the rate and amplitude of which depend on the fundamental nature of the protein–NA interaction as discussed below.

### NA–protein binding kinetics

The simplest protein–NA interaction to characterize is one in which the protein has only one binding conformation (Figure 2*a*). Specifically, each bound protein has the same binding site size (occludes the same number of nts). In this case, the ssNA substrate can be modeled as an array of binding sites, where each site acts independently and can occupy one of two states, protein-free and protein-bound.

$$\text{unbound} \underset{k_{off}}{\overset{ck_{on}}{\rightleftharpoons}} \text{bound} \tag{2}$$

Transition rates between these two states depend on the specific protein–NA interaction (e.g. different proteins have different binding affinities), but the rates are equivalent for each identical binding site. The rate of binding (of unbound sites becoming bound) is dependent on the concentration of free protein ($c$) and is denoted as $ck_{on}$, while the rate of dissociation (of bound sites becoming unbound) is concentration-independent and is denoted as $k_{off}$. Zooming out, the degree of protein saturation ($\theta$) for the entire

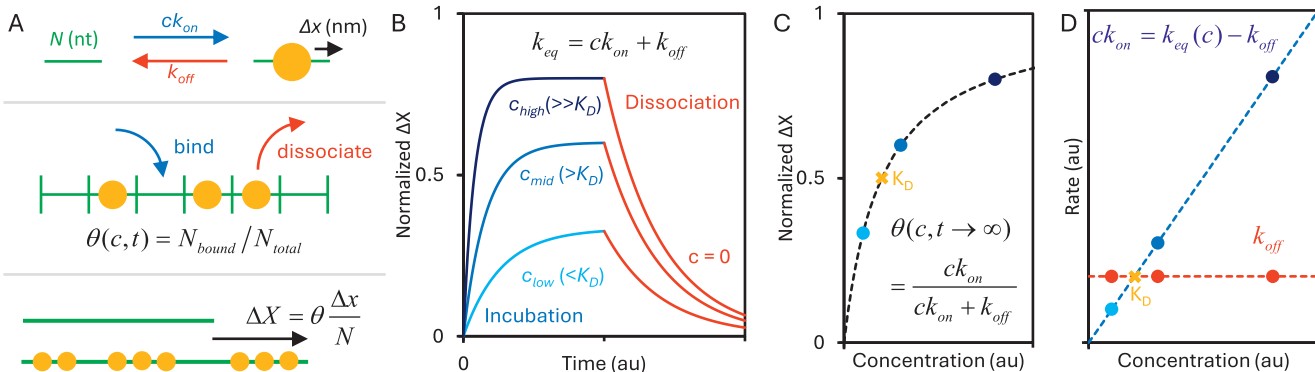

**Figure 2.** Simple bimolecular, reversible binding. (a) An NA substrate is modeled as an array of a fixed number of binding sites. Each site is protein-free or occupied, and the fraction of sites occupied depends on the free protein concentration and the time after the protein is introduced into the system. Each binding site is the same length ($N$) and acts independently. Empty sites are filled by proteins at a concentration-dependent on-rate and bound proteins leave at a fixed off-rate. The effective length of each binding site is changed by a fixed length ($\Delta x$) when protein is bound so that the length difference between an entire protein-free and protein-saturated substrate is the length change divided by the binding site size. (b) When protein is added to the sample (blue), the ssDNA extension changes at an observed rate equal to the sum of the on and off rates, and the total extension change reached at equilibrium increases with protein concentration. When free protein is removed, the ssDNA returns to its protein-free state. (c) The equilibrium ssDNA extension change as a function of protein concentration forms a binding isotherm, with the protein concentration that results in half as much change as seen at full saturation equal to $K_D$, by definition. (d) The rate of binding increases proportionally with protein concentration while the rate of dissociation is constant. The two rates intersect at $K_D$.

NA substrate can be simply calculated as the number of binding sites occupied by a protein divided by the total number of binding sites.

$$\theta(c,t) = n_{\text{bound}}/n_{\text{total}} \tag{3}$$

This value will vary with free protein concentration and over time if the system is not in equilibrium. The OT system does not directly measure protein binding, but rather NA conformation (the substrate's change in extension, $\Delta X$). Note, that the measured NA length ($X$) and change in length ($\Delta X$), are typically reported in normalized units of nm/nt, where the absolute length in nm is divided by the substrate length in nt. This allows results to be compared and interpreted consistently regardless of the specific substrate used in experiments, which is typically arbitrary and determined by (commercial) availability or technical constraints rather than a specific biological function. For example, dsDNA consistently has a normalized contour length of 0.34 nm/nt for both commercially available and widely used lambda phage DNA (48.5 kbp) and plasmid pUC19 (2.7 kbp), even though their absolute extensions vary by more than an order of magnitude.

If only one protein binding conformation is possible, then the extension change and degree of protein saturation are directly proportional.

$$\Delta X = \theta \frac{\Delta x}{N} \tag{4}$$

The maximal extension change, which is achieved when the NA is fully protein-saturated, can be experimentally determined by titrating protein concentration (at sufficiently high protein concentration, the amplitude of the extension change will asymptote, indicating saturation). If all proteins bind in the same conformation, then each binding site (size $N$ nt) will have its extension altered by a constant value $\Delta x$.

OT systems allow for changing free protein concentration around the isolated NA molecule, either by flowing in different buffers into the sample while keeping the trap stationary or by moving the trap into a different location within the sample. This drives the system out of equilibrium, which reveals the kinetics of the protein–NA interaction. When protein-free NA is suddenly introduced to free protein

(incubation), the extension change over time takes the form of an exponential decay (Figure 2b).

$$\Delta X(t) = \Delta X_{\text{eq}}\left(1 - e^{-k_{\text{eq}}t}\right) \tag{5}$$

The rate constant of equilibration ($k_{\text{eq}}$) is the sum of the two fundamental rates of binding ($k_{\text{on}}$) and dissociation ($k_{\text{off}}$).

$$k_{\text{eq}} = ck_{\text{on}} + k_{\text{off}} \tag{6}$$

Thus, higher protein concentrations ($c$) result in the NA substrate reaching its equilibrium extension on a shorter timescale. $k_{\text{off}}$ can be easily isolated by measuring the rate of dissociation when free protein is removed from the sample ($c = 0$). In contrast, $k_{\text{on}}$ can be calculated from a single incubation–dissociation cycle by subtracting the dissociation rate (measured after protein is removed) from the equilibration rate (measured during protein incubation) and dividing by the protein concentration. Alternatively, the equilibration rate can be measured for several protein concentrations and then linearly fit, with the zero-concentration intercept and the concentration proportional slope yielding $k_{\text{off}}$ and $k_{\text{on}}$, respectively.

Additionally, the amplitude of the extension change at equilibrium increases with protein concentration (Figure 2c), as the extension change of the NA is directly proportional to the degree of protein saturation (Eq. (4)). The fraction of binding sites occupied at equilibrium ($\theta_{\text{eq}}$) can be written in terms of the fundamental rates of binding and dissociation, or in relation to the effective dissociation constant ($K_D$), which indicates the protein concentration at which the rates of binding and dissociation are equal, and equivalently, half of the binding sites are protein-occupied (Figure 2d).

$$\theta_{\text{eq}} = \frac{ck_{\text{on}}}{ck_{\text{on}} + k_{\text{off}}} = \frac{c}{c + K_D} \tag{7}$$

The degree of protein saturation can also be directly measured based on the amplitude of the extension change at equilibrium, where the extension before protein introduction indicates $\theta_{eq} = 0$ and the maximum extension change approached at the highest protein concentrations signifies $\theta_{eq} = 1$. Agreement between calculations of these fundamental parameters using either rate or amplitude measurements confirms that the system follows a simple two-

state mechanism, governed by bi-molecular binding. Thus, when agreement is not found, such as for the three different protein systems explored below, the possibility of proteins binding in more than one conformation must be explored.

## Single-stranded binding proteins

Single-stranded binding proteins (SSBs) are highly abundant proteins that have been identified in all domains of life, including viruses, and eukaryotic and prokaryotic cell nuclei. All SSBs have ssDNA (or ssRNA) binding grooves that do not discriminate between the ssNA sequences but do not bind double-stranded (ds) NA regions. SSBs are generally involved in DNA (and in some viruses RNA) replication, repair, and recombination. SSBs are able to promptly engage all available ssDNA (and ssRNA) templates, thereby protecting them from degradation by nucleases, as well as eliminating NA secondary structure. SSBs are known to facilitate all NA metabolic processes, most notably, the rate of DNA replication by the polymerase complex. As an extremely important class of proteins, SSBs from many organisms have been studied extensively over the past several decades, with much information on their structure and ssDNA binding modes accumulated. Despite their similar roles in various organisms from bacteria to humans, SSBs were found to be surprisingly diverse in their structure, ssDNA binding modes, and binding cooperativity. Most importantly, despite decades of research, it remains unclear: (i) how variable amounts of ssDNA template always remain protected by the variable bulk concentrations of SSBs during DNA processing and (ii) how strongly, and often cooperatively, bound SSBs are able to promptly dissociate from ssDNA to clear the way for the rapidly moving polymerase complex as it synthesizes the complementary DNA strand with rates of ~100–1000 nt/s. Many SSBs have C-terminal unstructured or poorly structured anionic tails that compete with ssDNA for binding to their SSB binding sites, while also serving as the attachment points for multiple cellular proteins that regulate ssDNA processing. It was suggested that the binding of C-terminal tails to these regulatory factors helps to promptly dissociate SSBs from ssDNA. However, no direct evidence of such activity was provided. Moreover, SSBs are also routinely used to improve the yields of *in vitro* PCR reactions where no such cellular cofactors are provided.

Despite commonalities in function, different SSBs need not be structurally similar. In this review, we specifically examine three SSB proteins, from various sources (bacteria, bacteriophage, and retrotransposon). Despite these differences, we will show that they all exhibit similar collective behaviors when examined in single-molecule experiments. Perhaps the most well-studied such protein is the SSB of *E. coli* (*Ec*SSB). *Ec*SSB is a homotetramer, with each 19 kDa monomer comprising an N-terminal domain (NTD) containing an oligonucleotide binding (OB) fold (the protein's ssDNA binding site/groove), a C-terminal domain (CTD) with a conserved 9-amino acidic tip, and a poorly conserved intrinsically disordered linker (IDL) (Antony et al., 2013; Kozlov et al., 2015; Raghunathan et al., 2000; Raghunathan et al., 1997; Tan et al., 2017). The OB domain contains both the high-affinity DNA binding grooves and interfaces for interprotein interactions responsible for stable tetramerization. *Ec*SSB can bind ssDNA in multiple conformations, typically identified by the total number of nt occluded, such that higher binding site size states wrap more NA substrate around the OB tetramer (Bujalowski & Lohman, 1986; Bujalowski et al., 1988; Lohman et al., 1988; Lohman & Overman, 1985; Lohman et al., 1986). Free protein concentration (or equivalently protein:nt ratio),

salt conditions, and template tension (for force spectroscopy studies) affect the occupancy of these distinct binding modes (Bujalowski & Lohman, 1986; Bujalowski et al., 1988; Kozlov et al., 2019; Lohman et al., 1988; Lohman & Overman, 1985; Lohman et al., 1986; Suksombat et al., 2015). Three wrapping states are typically observed for *Ec*SSB bound to ssDNA in the absence of applied tension, with binding site sizes of 65, 56, and 35 nt. However, OT experiments observing single proteins binding to a 70 nt ssDNA substrate identified binding of as little as 17 nt by an individual tetramer under increased tension (Suksombat et al., 2015).

A model of the path through which a ~65 nt ssDNA substrate accesses the binding grooves of all four OB domains has been established based on X-ray crystallographic structural data (Raghunathan et al., 1997). While the exact topologies of other binding modes have not been structurally resolved, they are generally consistent with a discrete number of the OB folds being occupied by ssDNA. Some experiments have observed evidence of ssDNA segments as short as 8 nt binding to *Ec*SSB, including sedimentation of 8 nt poly dT oligos with a stoichiometry of more than 3 oligos per tetramer (Krauss et al., 1981), the addition of a poly dT ssDNA overhang to a hairpin increases protein-mediated hairpin destabilization, (Grieb et al., 2017), and AFM observation of *Ec*SSB localization to 8 nt poly dT overhangs at the end of dsDNA substrates (Naufer et al., 2021).

Even when NA-bound, *Ec*SSB is highly dynamic. Single-molecule FRET experiments measured a dynamic equilibrium between structural states (Roy et al., 2007), and diffusion of wrapped protein along its ssDNA substrate (Roy et al., 2009). Fluorescent imaging of *Ec*SSB-ssDNA complexes has even resolved multiple sequential kinetic steps, from measurements of the concentration-dependent rate of free protein binding (Kozlov & Lohman, 2002b) and the concentration-independent rate of wrapping (Kuznetsov et al., 2006), to the slow addition of additional protein to an *Ec*SSB-occupied substrate (Kunzelmann et al., 2010) and direct transfer of an *Ec*SSB tetramer between two different ssDNA substrates (Kozlov & Lohman, 2002a).

The depth of research on *Ec*SSB makes it a good model to compare with other less well-defined protein systems. The results we observe using optical tweezers (Naufer et al., 2021) can be directly related to previous research using different experimental systems. Since then, we have also observed significant commonalities with how both the gene 32 protein of T4 bacteriophage (T4 gp32) (Cashen et al., 2023; Cashen et al., 2024a) and the open reading frame (ORF) 1 protein of the LINE 1 retrotransposon (L1-ORF1p) (Cashen et al., 2022; Cashen et al., 2024b) interact with an ssNA substrate. As we will discuss further, the known framework in which *Ec*SSB interacts with ssDNA in multiple binding conformations can be generalized to explain both our experimental data and the function of these other proteins.

T4 gp32 is a 33.5 kDa monomer comprising three domains: a central ssDNA binding core (residues 22–253), a positively charged NTD (residues 1–21), and a negatively charged CTD (residues 254–301) (Karpel, 1990). The gp32 core domain binds ssDNA (7 nt occluded site size) in a small, positively charged cleft created by a single OB-fold, conferring the protein with largely sequence-independent binding and the ability to effectively discriminate against duplexed dsDNA (Shamoo et al., 1995; Theobald et al., 2003; Wu et al., 1999). gp32 forms highly stable protein filaments on ssDNA mediated by cooperative interprotein interactions between the NTD of a nucleic acid-bound monomer and the core domain of an adjacently bound protein (Casas-Finet et al., 1992;

Lonberg et al., 1981). Previous light scattering and circular dichroism measurements suggested that NA-bound gp32 filaments wind ssDNA, resulting in a relatively stiff, helical protein-DNA structure (Kuil et al., 1990; Kuil et al., 1988; Scheerhagen et al., 1989; Scheerhagen et al., 1985a; Scheerhagen et al., 1985b; van Amerongen et al., 1990). These findings were recapitulated by recent single-molecule DNA stretching experiments, which showed that cooperatively bound gp32 simultaneously rigidifies and compacts ssDNA, characterized by its increased persistence length and reduced contour length, respectively (Cashen et al., 2023), an expected signature of a helical protein filament (Griffith & Formosa, 1985; Lee et al., 2004; Takahashi & Norden, 1994; Wu et al., 2004; Xu et al., 2017; Yang et al., 2001; Yu et al., 2004). The gp32 CTD, on the other hand, is believed to primarily help coordinate DNA replication via direct (heterotypic) interactions with other constituents of the T4 replisome (Alberts & Frey, 1970a; Alberts & Frey, 1970b; Krassa et al., 1991; Lefebvre et al., 1999; Morrical et al., 1996; Nelson et al., 2008) while also competing with the ssDNA for the central domain's binding cleft, thereby moderating the strength of the individual protein ssDNA interactions in a salt-dependent manner (Pant et al., 2005).

The exact structural details of the gp32-ssDNA complex remain incomplete. An initial X-ray crystal structure of the gp32 core (ssDNA binding) domain complexed to a $dT_6$ oligonucleotide revealed only weak electron density for the ssDNA lattice bound within the protein's OB-fold (Shamoo et al., 1995), suggesting that ssDNA is fairly mobile within the gp32 binding groove, allowing the protein to freely translocate (slide) along ssDNA (Jose et al., 2015a; Lee et al., 2016; Lohman & Kowalczykowski, 1981). However, the authors modeled four nucleotides of the $dT_6$ chain into the gp32 binding cleft, and the resulting structure suggested that at least two nucleotides were tightly bound within the protein's core. A more recent low-resolution crystal structure of gp32 in complex with the T4 Dda helicase and a $dT_{17}$ oligo (He et al., 2024) further defined the entire ssDNA binding surface of the gp32 monomer, as well as its interaction with Dda. Consistent with these structural studies, oligonucleotide-based binding measurements using proteolysis and DNA $T_m$ depression methods demonstrated that at least 2–3 adjacent phosphodiester bonds are required for gp32-ssDNA binding (Wu et al., 1999). However, this study also showed an increase in gp32-ssDNA binding affinity when the oligos were increased in length from 5 to 8 nt, suggesting that the number of interactive residues within the core may be variable and dependent on substrate length.

Bulk studies of gp32 binding have often utilized relatively short ssDNA substrates, limiting measurements of gp32-ssDNA dynamics to either single noncontiguous monomers or small clusters thereof (Camel et al., 2021; Jose et al., 2015a; Jose et al., 2015b; Lee et al., 2016; Lohman & Kowalczykowski, 1981). However, the length of a typical Okazaki fragment in T4-infected *E. coli* is 1000–2000 nt (Maloy & Hughes, 2013) (i.e., can accommodate ~150–300 proteins), indicating that greater ssDNA lengths are likely needed for a complete understanding of gp32 filament structure and organizational dynamics *in vivo*. In this regard, single-molecule DNA stretching experiments are able to extend our understanding of gp32 behavior by probing its binding to long ssDNA substrates, which may accommodate >1000 proteins. Previous measurements on force-melted λ-phage DNA revealed how competing interactions of the acidic CTD for access to the protein's OB-groove regulate its salt-dependent binding to ssDNA (Pant et al., 2004; Pant et al., 2005; Pant et al., 2003; Rouzina et al., 2005). These studies also helped explain the origin of the "kinetic block" to

dsDNA helix-destabilization (melting) by full-length gp32 that was observed in thermal melting experiments.

Similar to *Ec*SSB, gp32 has the seemingly paradoxical requirement to both stably bind and protect regions of ssDNA transiently exposed during replication while also ensuring their rapid release upon synthesis of the complementary strand. While gp32's high-affinity binding facilitates efficient coating of the discrete Okazaki fragments, such stable and highly cooperative binding could prevent the protein from being easily displaced from its ssDNA template. Indeed, previous stopped-flow measurements revealed that gp32 primarily dissociates from the ends of its cooperative clusters and that the rate of unbinding is too slow to account for the observed rate of DNA synthesis by T4 polymerase (Lohman, 1984a; Lohman, 1984b). Efficient protein recycling during DNA replication remains an important, open question for all SSBs.

LINE-1 (L1) is an intragenomic parasitic DNA element, comprising ~20% of the human genome, that amplifies within its host through a "copy-paste" mechanism known as retrotransposition (Furano, 2000; Goodier et al., 2013; Kazazian & Moran, 2017; Lander et al., 2001). L1 encodes two proteins, ORF1p and ORF2p, which assemble on their encoding transcript (*cis* preference) to form the L1 ribonucleoprotein (RNP), an essential intermediate of retrotransposition (Doucet et al., 2010; Howell & Usdin, 1997; Kulpa & Moran, 2005; Kulpa & Moran, 2006; Martin, 1991; Martin, 2010; Moran et al., 1996; Sahakyan et al., 2017). ORF2p provides reverse transcriptase and endonuclease activity (Feng et al., 1996; Luan et al., 1993; Mathias et al., 1991; Miller et al., 2021; Moran et al., 1996; Thawani et al., 2024). ORF1p, the major component of the L1 RNP, is a homotrimeric phosphoprotein that binds single-stranded nucleic acid (ssNA) nonspecifically with high affinity and exhibits NA chaperone activity (i.e., facilitates annealing and exchange of NA strands).

ORF1p contains a 51 amino acid intrinsically disordered NTD, which harbors two highly conserved phosphorylation sites necessary for retrotransposition (Cook et al., 2015; Furano & Cook, 2016), followed by a 14-heptad coiled-coil (CC), which mediates the trimerization of ORF1p monomers (Boissinot & Sookdeo, 2016; Callahan et al., 2012; Khazina et al., 2011; Khazina & Weichenrieder, 2018; Martin et al., 2003). The ORF1p coiled coil is evolutionarily labile (subject to rampant amino acid substitutions) (Furano et al., 2020). However, despite such variability, mutational analysis has shown that ORF1p activity is exquisitely sensitive to its CC sequence (Adney et al., 2019; Goodier et al., 2007; Naufer et al., 2016), suggesting that the persistence of L1 activity requires periodic remodeling of the ORF1p coiled coil. In contrast, the carboxy-terminal half is highly conserved and comprises two domains: a noncanonical RNA recognition motif (RRM) (Khazina & Weichenrieder, 2009), which contains two additional phosphorylation sites required for retrotransposition, and a CTD, which terminates in a 46 amino acid intrinsically disordered sequence.

Residues within the RRM and CTD endow ORF1p with high-affinity ssNA binding and NA chaperone activity *in vitro*. However, these properties are only evident in the context of the trimer (Basame et al., 2006; Callahan et al., 2012; Januszyk et al., 2007; Khazina et al., 2011; Khazina & Weichenrieder, 2009; Kolosha & Martin, 2003; Kulpa & Moran, 2005; Martin, 2010; Martin et al., 2003; Martin et al., 2008; Martin et al., 2005; Martin et al., 2000; Moran et al., 1996). Mutations in the RRM or CTD domains that eliminate NA chaperone activity also abolish retrotransposition, suggesting a primary role of ORF1p chaperone activity in L1 replication (Martin et al., 2005). However, the mechanistic details of this activity are not known. FRET-based assays showed that

ORF1p can stabilize mismatched oligonucleotide duplexes (Callahan et al., 2012), which are likely to be encountered during the hybridization of the target-site DNA and L1 transcript to generate a productive primer for cDNA synthesis by ORF2p.

## Review of single-molecule experiments

### Measuring multimode binding

Recently published work using similar OT single molecule techniques for three different protein systems observed strong evidence that more than one binding state must be present (Cashen et al., 2023; Cashen et al., 2024a; Cashen et al., 2022; Cashen et al., 2024b; Naufer et al., 2021). In particular, incubation experiments show a non-monotonic extension change response, both in terms of extension over time and equilibrium extension change (Figure 3a–c). Low protein concentrations (order of 1 nM) result in incubation curves that resemble simple bimolecular binding, as the ssDNA's extension decreases monotonically before approaching a highly compact equilibrium. However, while increasing protein concentration does increase the rate of initial compaction, the final equilibrium compaction is reduced as the extension begins to increase over time upon reaching a minimum value. Thus, to analyze the kinetics of the incubation process, the data must be split into two regimes (Figure 3d). First, the ssDNA extension decreases as the binding proteins wrap and compact the substrate. Eventually, the ssDNA must be sufficiently

saturated such that in order to accommodate additional protein, already bound protein must decrease its binding site size by switching to a less wrapped, and less compacted, state. As a result, both rates (initial compaction and subsequent elongation) increase with free protein concentration, but the secondary binding rate is an order of magnitude slower. This discrepancy in rate can be explained by the energetic barrier of partially unwrapping already bound protein to accommodate additional protein into the saturated complex, which is not a necessary step when the ssNA is initially protein-free. The equilibrium ssDNA extension at equilibrium can also be interpreted as a competition between a more and a less compacted binding state by extending Eq. (2) into a two-step, three-state reaction (Naufer et al., 2021).

$$\theta_0 \underset{k_{off}}{\overset{ck_{on}}{\rightleftharpoons}} \theta_u \underset{k_u}{\overset{k_w}{\rightleftharpoons}} \theta_w \qquad (8)$$

In this scheme, the binding and wrapping of ssNA by protein are separated into two distinct steps. If the maximally and minimally compacted ssDNA extensions observed are associated with full occupancy of the wrapped ($\theta_w$) and unwrapped state ($\theta_u$), respectively, then the occupancy of both states can be calculated for any intermediate extension through interpolation (Figure 3e). Such analysis shows a smooth transition between the two states as a function of protein concentration that can be reproduced by simulating the three-state reaction in Eq. (8) or by approximating the interconversion between the states as a simple binding isotherm.

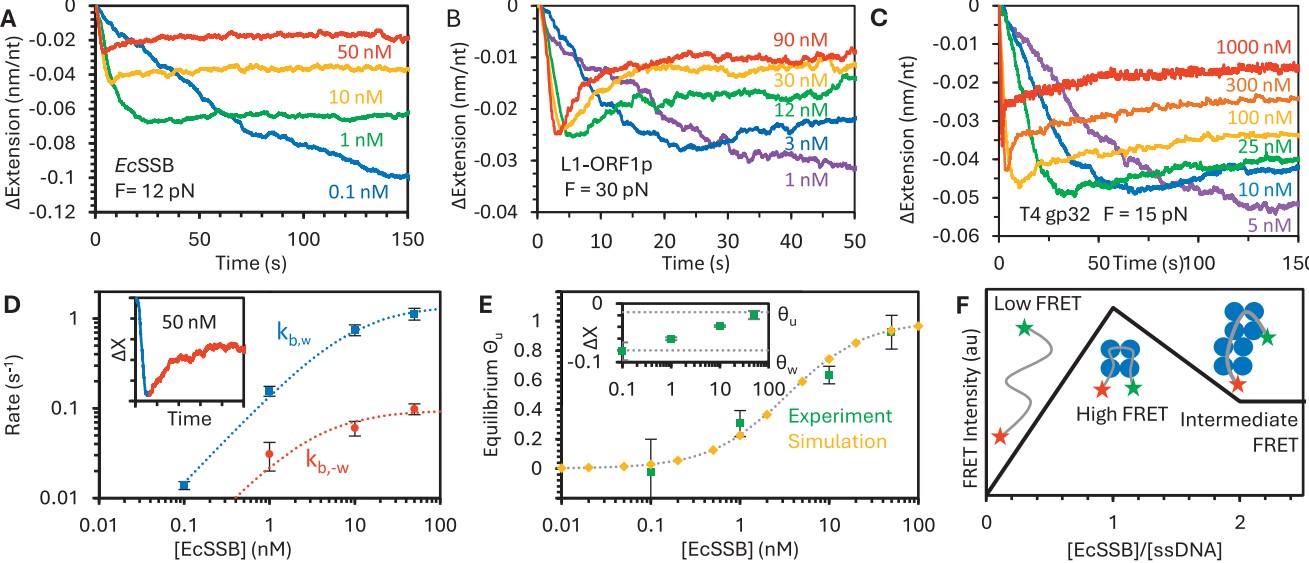

**Figure 3.** Non-monotonic response to protein concentration. (a) An ssDNA molecule is held at constant force (12 pN) while *Ec*SSB is flowed into the sample, which binds and wraps the ssDNA, resulting in a decrease in extension. At low concentration (0.1 nM), the extension decreases monotonically, exponentially approaching an equilibrium value, consistent with simple concentration-dependent diffusion limited bi-molecular binding. Increasing protein concentration increases the initial rate of extension change, but ssDNA extension does not continue to decrease to the same value, and instead abruptly begins extending, equilibrating to a less compact conformation. This biphasic pattern is consistent with the protein initially wrapping the substrate to maximize binding site-NA contacts (largest binding site size), and then partially unwrapping to decrease binding site size and allow for additional protein binding. The same trend is observed for L1-ORF1p (b) and T4 gp32 (c). Note, that while the exact force used in experiments alters the absolute extension changes and kinetic rates, this trend is observed over a range of forces for each protein system. (d) The initial rate of ssDNA compaction (blue) and the secondary rate of ssDNA elongation (red) both increase with protein concentration (reflecting diffusion-limited bi-molecular binding), but the second step rate is an order of magnitude slower due to additional protein having to compete with and partially displace already bound protein. The inset shows 50 nM data from panel A with two phases marked. (e) The equilibrium extension change as a function of *Ec*SSB concentration (inset) is converted to a measure of occupancy of the maximally and minimally compacted wrapping states (green squares). The occupancy of the unwrapped state ($\theta_u$) increases with protein concentration, similar to a standard binding isotherm (dotted line), and can also be reproduced by numerically simulating a three-state model (yellow diamonds). (f) An additional experimental system that directly measures NA conformation, rather than protein binding itself, is FRET measurement with coupled dyes located at either end of the binding substrate (Roy et al., 2007). When the pair of dyes are separated by a free 70 nt long poly dT ssDNA, little FRET intensity is observed. When the ssDNA is bound by a single *Ec*SSB tetramer, the exact structure of the ssDNA wrapping places the labels in close proximity, resulting in high FRET intensity. Increasing the protein to ssDNA ratio to above 1:1, however, results in two tetramers simultaneously binding the substrate, each in a reduced binding site size state (35 nt), moving the dyes further from each other and decreasing FRET efficiency to an intermediate value.

The analysis shown here for *Ec*SSB (Naufer et al., 2021) was also used to separate the multiple binding states and kinetic steps for L1-ORF1p (Cashen et al., 2022; Cashen et al., 2024b) and T4 gp32 (Cashen et al., 2023; Cashen et al., 2024a). One advantage of using EcSSB as a model system is that its ability to interconvert between different wrapping states with different binding site sizes has been well-documented using many different experimental assays. For example, one assay that also directly measures the conformation of an NA substrate (rather than directly measuring protein bound) consists of a FRET pair of dyes on both ends of a binding substrate (Roy et al., 2007)(Figure 3f). When no protein is present, the ends are effectively uncoupled, leading to low FRET intensity. For *Ec*SSB, when one protein tetramer binds a 70 dT substrate, the two ends are brought together, resulting in high FRET efficiency. In contrast, if two tetramers bind the ssNA simultaneously (each occupying 35 nt), the resulting structure places the two dyes further apart. As a result, EcSSB titration experiments with this assay also return a non-monotonic response with concentration, similar to the OT experiments described above. In contrast, any experiment that directly measures bound protein (such as directly fluorescently labeling the protein) will instead simply measure a continuous increase in signal as free protein concentration is increased, obfuscating conformational transitions of the protein–NA complex.

### Evidence of facilitated dissociation

Removing protein from the sample allows for the direct observation of protein dissociation from the ssNA substrate. One issue with working with proteins with very strong binding affinity, however, is that the binding to the substrate may be too stable to observe measurable dissociation on experimental timescales. For *Ec*SSB binding to ssDNA held at low force, removing free protein does not result in significant ssDNA extension change (Naufer et al., 2021) (Figure 4a). Subsequently increasing the EcSSB concentration, however, does result in a change in ssDNA extension. Compaction is reduced, and the ssDNA equilibrates to the same length as when the substrate is initially incubated with the same high protein concentration. In contrast to the equilibrium reached when the ssDNA is incubated with low *Ec*SSB concentration, this less compact state is very unstable, and removing free protein at this point causes the ssDNA to quickly recompact. After reaching the same highly compact state observed during the initial incubation with low protein concentration, the system becomes stable again and no further extension change is observed. These result demonstrate that even under conditions where *Ec*SSB does not fully dissociate (protein-free ssDNA is never recovered), the protein-NA complex must be able to rapidly reorganize based on the local density of protein present.

If the ssDNA tension is increased, binding is further destabilized such that full protein dissociation can be observed. That is, eventually the ssDNA returns to its original protein-free conformation (Figure 4b). However, even under these conditions, the ssDNA-EcSSB complex first rapidly recompacts when free protein is removed before slowly elongating as the rest of the protein dissociates. The same effect is again observed for L1-ORF1p (Cashen et al., 2022; Cashen et al., 2024b) (Figure 4c) and T4 gp32 as well (Cashen et al., 2023; Cashen et al., 2024a) (Figure 4d).

Similar to the incubation experiments, the dissociation data must be split into two distinct regimes. The rapid reorganization that results from protein re-wrapping the ssDNA when excess

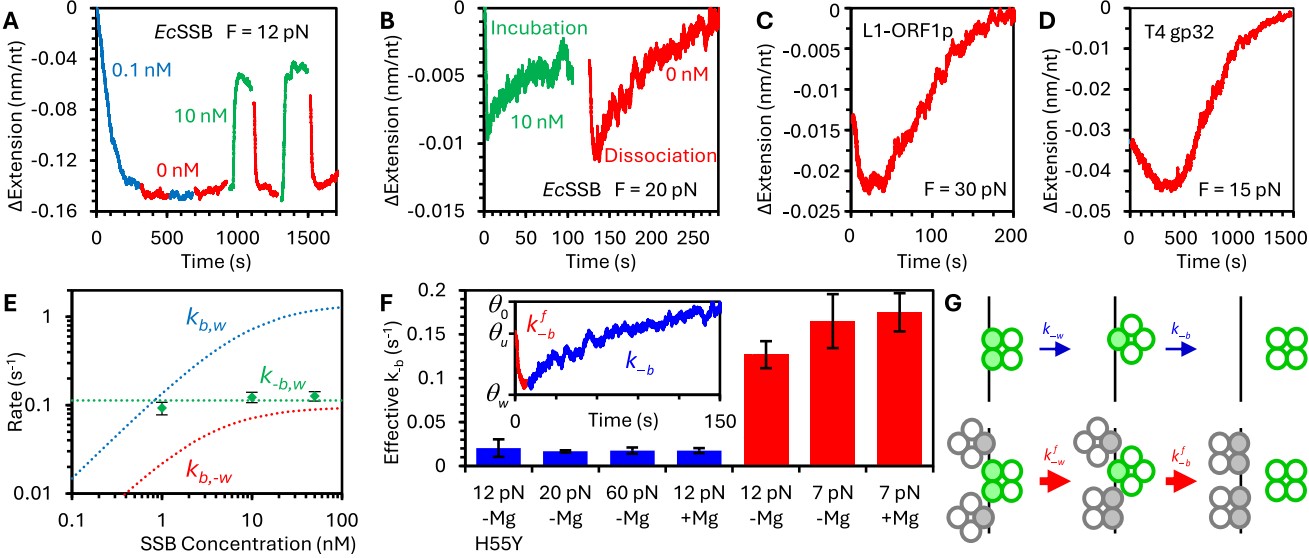

**Figure 4.** Facilitated dissociation of over-saturated protein. (a) After ssDNA has been incubated with protein, the free protein concentration is suddenly changed to measure the re-equilibration of the complex. At low forces, *Ec*SSB binding is very stable, and little dissociation is observed on the ~100 s timescale. However, subsequently, increasing protein concentration reduces the decrease in extension, similar to when the ssDNA is initially incubated with high protein concentration (Figure 3a). When free protein is removed after oversaturation, the ssDNA rapidly recompacts, consistent with facilitated dissociation of excess protein to allow remaining bound proteins to return to their most wrapped, stably bound state. (b) At higher forces, *Ec*SSB wrapping is destabilized, allowing measurable protein dissociation, as the ssDNA returns to its protein-free state. However, the initial dissociation of excess (oversaturating) protein occurs much faster than the final full dissociation, consistent with a mechanism of facilitated dissociation. Similar trends are observed for L1-ORF1p (c) and T4 gp32 (d). (e) The rate of *Ec*SSB dissociation that results in further wrapping of the ssDNA is constant with respect to protein concentration during incubation. Compared to the rates of binding, the dissociation rate crosses the initial binding rate at <1 nM but only approaches the secondary binding rate at >10 nM, consistent with the ssDNA becoming saturated at low protein concentration but requiring much higher concentrations to oversaturate. (f) While full protein dissociation is enhanced by increased force, salt, or certain protein mutations, this rate remains an order of magnitude slower than the fast initial rate of facilitated protein dissociation. (g) Full protein dissociation (top) is slow due to the energetic barrier of removing binding energy between the ssDNA substrate and the binding domains of *Ec*SSB. For oversaturated substrates (bottom), binding contacts released during dissociation are replaced by other bound proteins, removing this energy barrier and facilitating dissociation.

protein is removed occurs at a constant rate regardless of the initial *Ec*SSB concentration during incubation (Figure 4*e*). For *Ec*SSB, when compared to the two rates of binding observed during incubation, this initial dissociation rate intersects the initial binding rate at <1 nM, consistent with low protein concentration able to fully saturate the ssDNA substrate. In contrast, the secondary binding rate asymptotes to the initial dissociation rate at a high concentration (>10 nM), consistent with the high *Ec*SSB concentration needed for the less wrapped state to become dominant. The secondary dissociation rate, where the ssDNA extends back to its original protein-free length, is much slower and is only detectable under conditions that augment full dissociation, such as destabilizing binding using excess applied force or high salt buffer that screens electrostatic binding interactions (Figure 4*f*). The order of magnitude difference between these two dissociation rates indicates that some biophysical process must be stimulating protein dissociation from this oversaturated state. This can be explained by the presence of multiple binding modes with different effective binding site sizes. If the ssDNA is protein oversaturated (bound protein is in its lower binding site size state to accommodate excess protein), then when a single protein dissociates, neighboring proteins can switch to the higher binding site size state, effectively absorbing the released NA substrate (Figure 4*g*). This process, in which the ssNA substrate released by a dissociating protein is reabsorbed by another protein, is referred to as facilitated dissociation (Erbaş & Marko, 2019). In contrast, when protein dissociates from an undersaturated substrate, protein-free NA is left behind. Thus, full dissociation has an additional energy barrier due to the loss of protein–NA biding energy, while facilitated dissociation is closer to an isoenergetic process in which the loss of the final ssNA contact before a protein is released from the substrate is replaced by the analogous interaction with a neighboring protein.

### *Protein structure and function*

The ssNA interactions exhibited by all three proteins discussed here can be related to the common features of their complexes with ssNA, despite clear differences in structure and multimerization (Figure 5*a*). *Ec*SSB and L1-ORF1p naturally form homotetramers and homotrimers, respectively. As a result, each homo-oligomer intrinsically has multiple binding domains. If each binding domain can bind ssNA semi-independently, with the substrate winding around the protein oligomer in different conformations to access the binding grooves of a discrete number of subunits, then the proteins inherently have the ability to alter their effective binding site size.

T4 gp32, conversely, is primarily monomeric in solution. However, interactions between the NTD and the core domains of neighboring proteins bound to an ssDNA substrate confer the protein with cooperative binding, resulting in the formation of long protein clusters on the substrate. Again, this results in many binding interfaces present on each homo-oligomeric filament, which must allow for the modulation of protein:ssDNA stoichiometry. Our results are consistent with the ssDNA substrate helically winding around the protein filament while remaining highly dynamic due to the ability to partially unwind to accommodate additional protein at high protein:ssDNA stoichiometries (Cashen et al., 2023). Moreover, we found that critical protein oversaturation resulted in filament unwinding such that the cooperative interprotein interactions largely vanished, enabling rapid protein displacement from across the entire ssDNA substrate, relieving oversaturation.

However, the structural differences between these proteins confer different biophysical interactions with the ssNA that are measurable using OT. First, L1-ORF1p differs from the other proteins in that also exhibits an RNA packaging function mediated by intertrimer interactions, in which the protein must stably bind and compact a copy of the L1 sequence. Correspondingly, when an L1-ORF1p saturated ssDNA is held at low force (≤5 pN), the substrate continues to compact over time (Figure 5*b*). Additionally, when left for increasingly long incubation times, high-force stretching of the ssDNA reveals permanent compaction. When WT L1-ORF1p is replaced with an inactive (retrotransposition-deficient) mosaic of modern and ancestral strains of L1, this secondary compaction ability is lost (Cashen et al., 2022). As T4 gp32

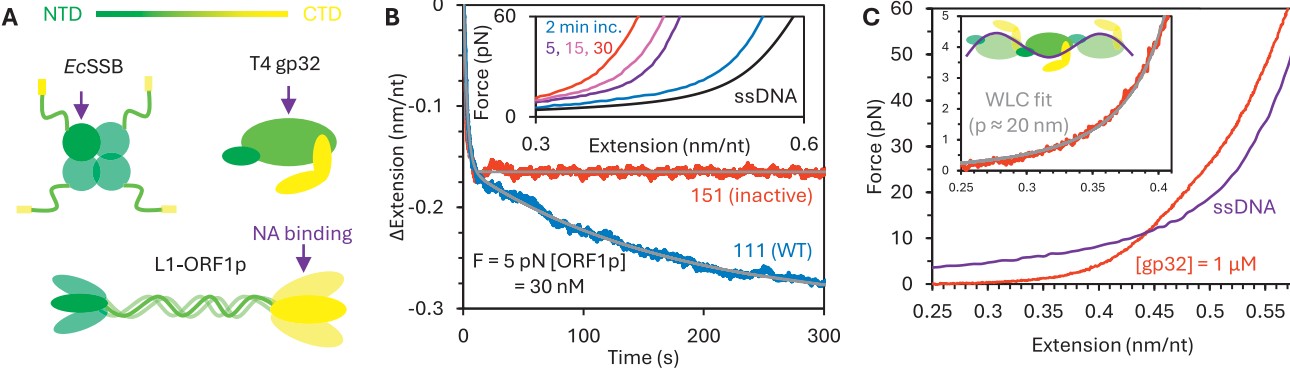

**Figure 5.** Comparison of protein structures. (a) *Ec*SSB is a homotetramer mediated by its oligonucleotide binding domains (NTD, green). Each subunit has an intrinsically disordered tail (green-yellow) ending with an acidic tip at the C-terminus (yellow). L1-ORF1p forms trimers through its central coiled coil domain, with NA binding activity located in the RRM-CTD. T4 gp32 is a monomer in solution, but its NTD binds to the core domain of a neighboring protein to enable cooperative binding and oligomerization of NA-bound proteins. The T4 gp32 CTD competes with NA for access to the core binding domain and modulates binding in a salt-dependent manner. Purple arrows indicate the NA binding domain of each protein. (b) At very low forces, when ssDNA is not straightened, L1-ORF1p exhibits a secondary compaction phase that further reduces ssDNA extension after protein saturation. Stretching the ssDNA–protein complex after increasing incubation times (inset) confirms a stable, non-reversible shortening of the substrate. Certain protein variants that are deficient in retrotransposition activity lack this secondary compaction function. As this tight compaction is effectively irreversible, it would seemingly interfere with the protein's SSB-like function during polymerization. However, tight NA compaction by L1-ORF1p is likely critical to its role in RNA packaging, a function not exhibited by the other studied proteins. (c) T4 gp32 forms long protein filaments on ssDNA, greatly reducing the force needed to straighten the substrate (or equivalently, increasing ssDNA extension at low force). Fitting the FEC of protein-saturated ssDNA to the WLC model returns an effective persistence length of ~20 nm, much longer than the length of a single protein, indicating that the interprotein interface is semi-rigid and preserves the relative orientation of neighboring proteins in the filament.

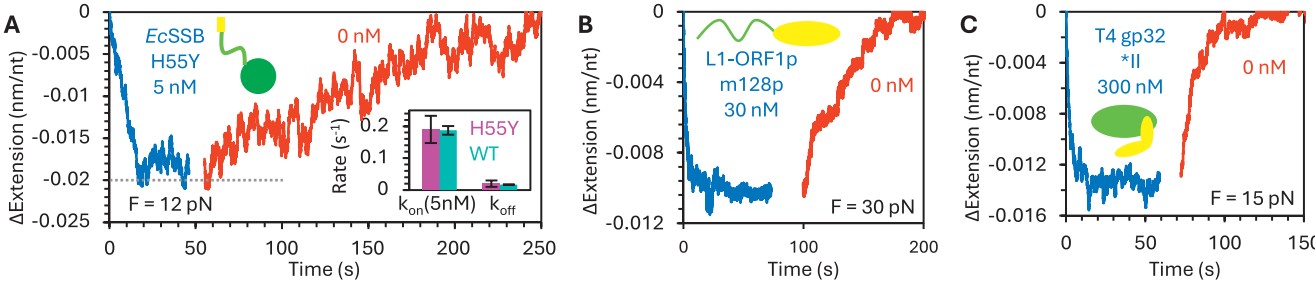

**Figure 6.** Monomeric, non-cooperative protein variants. (a) ssDNA is incubated with H55Y mutant *Ec*SSB, which does not form protein tetramers. ssDNA extension decreases monotonically during incubation and increases monotonically when free protein is removed, returning to its initial protein-free state. The amplitude of compaction at saturation (gray dotted line) matches the extension of ssDNA over-saturated with WT protein and the rates of binding and dissociation match the rates of initial binding and full dissociation of the WT protein (inset). Monomeric protein variants m128p (truncated at 128[th] residue) L1-ORF1p (b) and *II (NTD removed) T4 gp32 (c) also display similar two-state, bimolecular, non-cooperative, reversible binding.

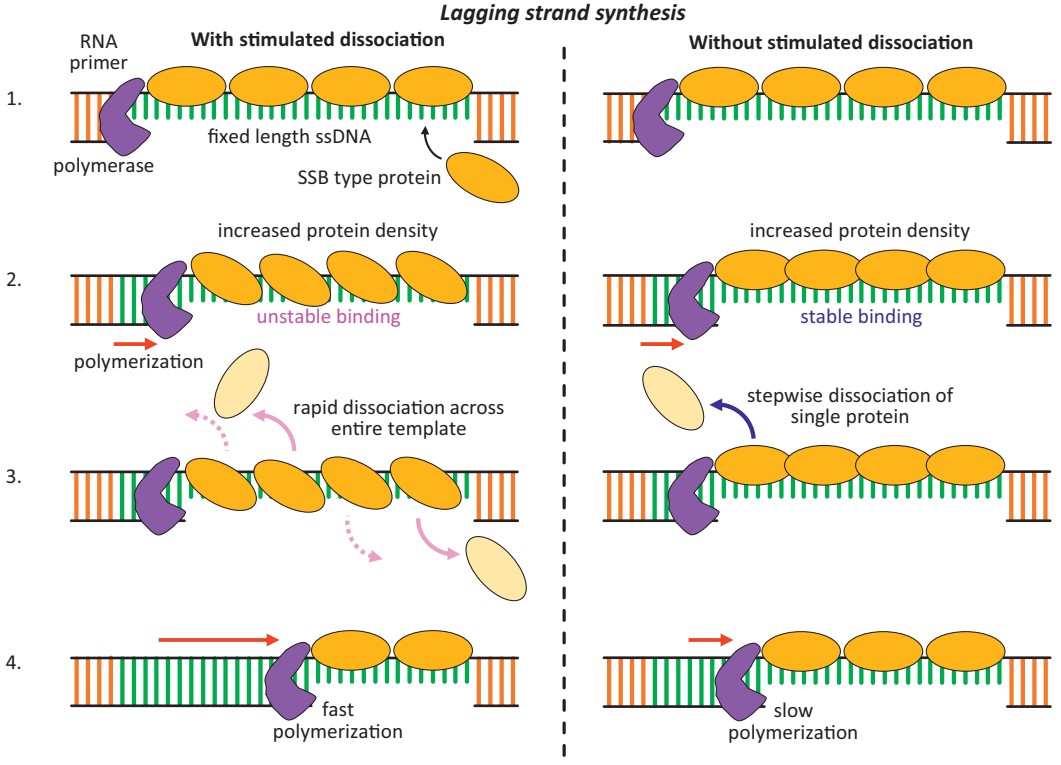

**Figure 7.** Biological impact of unstable, oversaturated protein binding state. A protein with high-affinity ssNA binding that is expressed in sufficient quantity will quickly saturate any transiently exposed ssNA regions that form during processes such as Okazaki fragments production during DNA replication. As a polymerase proceeds along the template strand, the ssNA region is reduced in length, effectively increasing protein density. If the proteins are able to switch to a lower binding site size and reduced binding free energy conformation (left side), facilitated dissociation is enabled in which any protein on the lattice can dissociate while neighboring proteins compensate for the loss of interaction free energy by absorbing any additional nucleotides released by switching back to a larger binding site size state. In contrast, if the protein remains in its most stable state (right side), in which the binding site size is maximized by utilizing the entire binding surface on each protein, then protein dissociation is inhibited. Instead, some other mechanism, such as direct interactions between the polymerase and the adjacent protein, must remove fully bound proteins in a stepwise, sequential manner, such that polymerization is rate limited by protein dissociation.

forms long, continuous protein filaments rather than discrete tetramers or trimers, the length scale of interprotein interaction is greatly increased relative to that observed for L1 ORF1p or *Ec*SSB. In addition to decreasing the contour length of the protein-saturated ssDNA substrate, a large (~30-fold) increase in persistence length is also observed (Figure 5c). ssDNA bound with T4 gp32 has an effective persistence length of approximately 20 nm, which is much longer than the length scale of the protein itself (Cashen et al., 2023). Thus, the orientation of neighboring proteins in the filament must

be preserved, resulting in the protein–ssNA complex behaving like a semirigid structure.

Our hypothesis that the dynamic multistate binding of these proteins is related to their ability to multimerize is supported by comparison experiments with non-multimerizing protein variants that exhibit neither oversaturated protein binding nor facilitated dissociation. Instead, simple single-state binding is recovered (Figure 6). First, a point mutation in *Ec*SSB's OB domain (H55Y) prevents the formation of homotetramers in

solution but does not inhibit the binding of the domain to ssDNA (Figure 6a) (Naufer et al., 2021). As a result, the binding of this monomeric mutant results in minimal ssDNA compaction, similar in amplitude to oversaturating WT protein conditions, where wrapping is destabilized to accommodate additional protein binding. Additionally, the binding is completely reversible, and the protein immediately begins dissociating when free protein is removed, extending the ssDNA back to its protein-free conformation. However, the rates of initial binding and full dissociation are still consistent with the rates observed for WT protein. Similarly, truncation of the L1-ORF1p protein at the 128th residue (m128p), removing the NTD and 10.5 heptads of the 14-heptad coiled coil, prevents trimer formation while leaving the binding domains in the RRM-CTD intact (Cashen et al., 2024b). Again, compaction is reduced and single-phased, and binding is reversible (Figure 6b). Finally, truncation of T4 gp32 to remove the entire NTD domain responsible for cooperative binding has the same effect (Figure 6c) (Cashen et al., 2023). Besides removing multistate binding, the large increase in persistence length is no longer observed, but is instead consistent with the length scale of a single binding site, indicating that neighboring proteins are no longer associated with a continuous filament, but instead act independently of one another.

### Biological function of variable conformation binding

The role of these proteins' multistate binding may be related to their need to perform the seemingly opposed functions of stably binding ssNA for protection while remaining dynamic enough to reorganize as NA processing proceeds. All three proteins bind lengths of ssNA that must be polymerized into ds form, ssDNA Okazaki fragments formed during lagging strand synthesis for *Ec*SSB and T4 gp32, and during replication of the L1 RNA transcript for L1 ORF1p. When first formed, the ssNA region has a discrete length and the limited pool of binding proteins must be sufficient to fully saturate this length, and all other ssNA regions present at a given time (Figure 7). Under these conditions, it is beneficial that ssNA binding proteins can occupy and occlude as many nts as possible. For example, by fully wrapping around all available domains, 65 nt ssNA (~35 nm of linear length) can be fully occluded by a single *Ec*SSB tetramer. However, for polymerization to proceed, these occluded nt must eventually be accessed by the polymerase enzyme. As the ssNA binding proteins do not naturally dissociate on a short timescale (as required by the protection function), the proteins must be removed by an active process. One candidate is a specific interaction between the polymerase and the binding proteins. Such interactions would be limited, however, to a single protein at the ds-ss junction and would have to proceed in a stepwise fashion. The ability of such strongly bound proteins to be removed one at a time in sufficiently rapid sequence as to allow efficient polymerization would appear difficult. However, the presence of multiple binding states alleviates this bottleneck. As polymerization proceeds, bound proteins can switch to a lower binding site state, giving the polymerase access to additional nts. Additionally, since these states have reduced contact with the ssNA substrate, binding is weakened, and dissociation is enabled for all proteins across the ssNA region. Finally, this mechanism also allows for facilitated dissociation, as observed in the OT experiments, where the dissociation of proteins from an oversaturated substrate is an order of magnitude faster than dissociation that leaves behind bare ssNA. Thus, the ssNA binding proteins can promptly reorganize and dissociate in front of the advancing polymerase so as to not delay DNA synthesis.

## Conclusions

Our OT experiments shed new light on the mechanism of prompt protein dissociation in front of a moving DNA polymerase. We observed that ssNA oversaturation (crowding) with any of the three studied ssNA binding proteins leads to rapid non-cooperative dissociation of excess protein from along the ssNA template. This mechanism is enabled by the structure of the proteins and the oligomers they form, in which multiple binding interfaces are present, allowing proteins to interconvert between stable and unstable conformations with distinct dissociation kinetics. Competition between these binding interfaces, in which the NA substrate released from one site can be reabsorbed by another empty site, enables rapid protein reorganization and facilitated dissociation.

**Open peer review.** To view the open peer review materials for this article, please visit http://doi.org/10.1017/qrd.2024.21.

**Data availability statement.** All data available in previously published manuscripts and upon reasonable request from M.C.W.

**Author contribution.** M.M., B.A.C., I.R. wrote the original manuscript. M.M., B.A.C., I.R., and M.C.W. reviewed and edited the manuscript.

**Financial support.** This work was supported by the National Science Foundation [MCB-1817712 to M.C.W.]

**Competing interest.** The authors declare no conflicts of interest.

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
