## [Reviewer Report]

This is a minireview about SSB-DNA binding from kinetic experiments with optical tweezers. The paper is quite dense for a general reader but might be useful for the specialists. I recommend publication after a few questions have been addeessed.

The espression DX=theta*Dx/N seems to me it misses a factor n equal to the total number of nucleotides, unless DX is defined as the change in extension per nucleotide.

What do you mean by oligonucleotide binding (OB) fold? Please explain.

Figure 1B shows predicted FECs of ssDNA molecules complexed with SSB. Do authors have data on this? I am pretty sure the experimental curves show a cooperative plateau for the binding-unbinding of SSB, something that is unpredicted in the FECs in figure 1C

What about sequence effects? Does SSB binds ssDNA in a sequence independent manner? I believe that the binding affinity and binding kinetics of SSB for dA, dT, dC and dG rich sequences is different. Could they discuss these effects?

The paper is dense for reading with figures containing many panels and long captions. It,would be good to select the most relevant results to leverage the content of the figures.

---

## [Reviewer Report]

The authors propose that optical tweezers (OT) experiments are useful for investigation of the binding mechanism of ssDNA binding proteins during DNA replication. The authors discuss the binding mechanism through studying examples for the three selected ssDNA binding proteins. Although the topic is interesting, there are some issues that should be clarified.

1. The manuscipt seems to have been submitted as a research article. However, there is no information about the experimental part.

2. The manuscript is hard to follow for the readers. There should be a distinct introduction part.

3. Some parts of the manuscript is too descriptive resembling a textbook. The information should be concise.

4. The manuscript should summarize the previous studies on the binding mechanism of ssDNA binding proteins. Novelty of the work should be emphasized.

5. It is not clear what the authors mean by DNA replicases. Is it polymerase or replicon?